# Multimodal Inplace Prompt Tuning for Open-set Object Detection

## ABSTRACT

The integration of large language models into open-world detection frameworks significantly improves versatility in new environments. Prompt representations derived from these models help establish classification boundaries for both base and novel categories within open-world detectors. However, we are the first to discover that directly fine-tuning language models in detection systems results in redundant attention patterns and leads to suboptimal prompt representations. In order to fully leverage the capabilities of large language models and augment prompt encoding for detection, this study introduces a redundancy assessment metric to identify uniform attention patterns. Furthermore, in areas with high redundancy, we incorporate multimodal inplace prompt tuning (MIPT) to enrich the text prompt with visual clues. Experimental results validate the efficacy of our MIPT framework, achieving a notable increase across benchmarks, e.g. elevating GLIP-L from 22.6% to 25.0% on ODinW-35, and 9.0% improvement on LVIS.

## CCS CONCEPTS

• **Computing methodologies** → **Object detection**.

## KEYWORDS

Open world detection, parameter efficient, multimodal learning

## 1 INTRODUCTION

Detection systems with large language models mark pivotal advancements in the open world [55]. As text prompts and visual regions are aligned through contrastive learning [17], new visual regions can be distinguished between classification edges shaped by linguistic world knowledge. This multimodal perception approach decouples the understanding of world knowledge from the only perspective of vision (e.g., traditional single-modal detection [10, 45]), facilitating the development of detection systems towards greater scale and enhanced capabilities.

Recent studies use two different methods to overcome the closed-vocabulary limitations. First, using fixed semantic representations from the frozen language model can make full use of the smoothness of the text space for categorizing objects [1, 44, 64]. The classification edge brought by the plain-text-pretrained large language model [4, 37], shows advances in the visual field. While innovative, such methods still have limitations in alignment for detection problems of long-tail and fine-grained categories. This is because alignment can only be unilaterally conducted from the visual backbone, but

the classification boundaries of the text still lack visual discriminativeness. Transitioning from frozen to tunable language models can offer higher flexibility. Finetuning allows the language model to adapt its pretrained representations to the specific visual domain [8, 61, 69]. Through closer alignment between text prompts and visual regions, language world knowledge recalibrated by visual contrast can lead to better prompt representation, thus providing more precise classification boundaries. Despite the pivotal role of language models, the nuances of how these models learn and adapt within such multimodal frameworks have been largely overlooked.

Initially, language models were designed to understand and generate contextualized texts. When integrated into detection systems [25, 34], text embeddings provide a novel view to define visual classification boundaries. Recent fully-finetuned detection models [25, 34, 67] concatenate categories into a template and finetune the language model to obtain the prompt representation of the textual template. However, our research reveals a degradation in the efficiency of the attention mechanisms for language models when directly fully-finetuned, as shown in the Section 4.2.3. This redundancy can result in model's inability to effectively encode the categories for rich semantics. Thus in turn, falls to detect such fine-grained or long-tail categories, as shown in the left column of Figure 1. Meanwhile, continuing full-scale fine-tuning on such categories would require substantial training costs. As depicted in the Figure 1, comparing to the full-finetuning (e.g GLIP-T [25] with more than 200 million parameters trainable), we aim to propose an efficient method that utilizes a small number of trainable parameters to improve the detection effectiveness for long-tail and fine-grained categories.

To this end, given the language model adaptation for detection that have been previously overlooked, we introduce a novel Multimodal Inplace Prompt Tuning Framework (MIPT) for open-set object detection. As shown in the right column of Figure 1, our MIPT is for the first time designed to identify redundancy within the weight distribution of the language model, and enrich the textual representation through a light-weight integration of visual exemplars. Consequently, text priors imbued with world knowledge are effectively augmented with visual cues, resulting in an enhanced classification boundary. Specifically, we introduce a Jensen-Shannon Redundancy evaluation to detect redundancy by assessing the variance of attention patterns intra- and inter-heads within transformer layers. Upon identifying redundant patterns, we apply multimodal inplace prompt tuning in these layers to refine textual prompts using visual cues, thereby augmenting the semantic depth of text encodings and improving model learning efficacy. Inplace tuning is achieved by performing parameter-efficient fine-tuning [15] for language models, thereby allowing it to reparameterizing internal encoding patterns without changing the original model structure. Furthermore, we develop cross-modal self-distillation to refine multimodal alignment between visual features and the augmented textual features in the fusion layers. Our experimental outcomes on GLIP-T, GLIP-L, and G-DINO-T showcase the remarkable efficacy of the

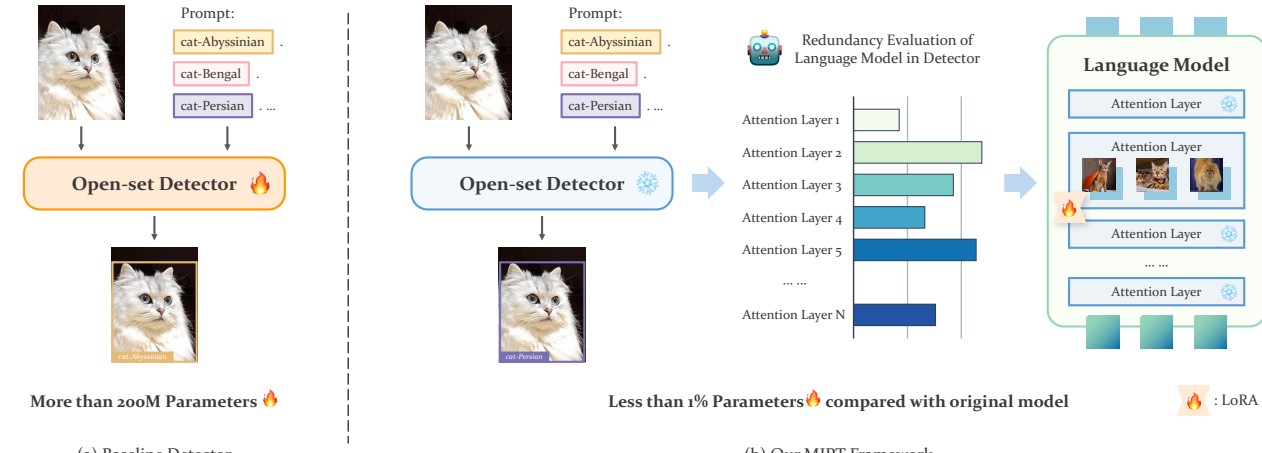

(a) Baseline Detector

(b) Our MIPT Framework

**Figure 1: A comparison of the baseline detector and our refined MIPT Framework. (a) Fully pretrained large detection models, such as GLIP, experience degradation in the language model, which makes encoding rich semantics for downstream categories challenging (especially failing to detect the fine-grained categories, such as various breeds of cats.). Resolving this issue through full parameter fine-tuning would entail substantial costs. (b) By diagnosing the redundant parts of the pretrained language model within the detector, we introduce visual clues into these redundant layers and employ inplace tuning with minimal parameters. This enhances text encoding and achieves precise fine-tuning with few samples for downstream tasks.**

MIPT framework, yielding substantial performance enhancements. Notably, GLIP-L (MIPT) demonstrates an impressive average precision improvement of 9.0% on the LVIS Val v1.0, reaching 25.0% on the ODinW-35, thus affirming our approach's capability to significantly elevate detection accuracy across diverse datasets.

Our contributions can be summarized as:

(1) We are the first to identify redundancy in attention distribution when integrating large language models into detection frameworks, which emphasizing the need for novel methods to improve fine-tuning and maximize their detection capabilities.

(2) We introduce the Jensen-Shannon Redundancy metric to efficiently evaluate and pinpoint attention pattern redundancies in language models during fine-tuning for detection tasks, enhancing optimization across transformer heads in multimodal settings.

(3) We propose Multimodal Inplace Prompt Tuning, a novel approach that recalibrates prompt representations using visual cues for language models. This adjustment is achieved without altering the original model structure, thus maintaining the integrity and generalizability of the model while significantly improving its performance in detection tasks.

## 2 RELATED WORK

### 2.1 Open-set Object Detection

The domain of open-set object detection has rapidly evolved with the integration of large language models and multimodal learning frameworks [19, 30, 54, 57, 58, 60, 66, 71, 73–76]. MDETR [17] introduces a model that aligns text phrases with visual regions

using a DETR-like architecture. This method improves object detection by using textual queries to guide the detection, resulting in a more versatile understanding of visual content. GenerateU [31], which aims to identify and name objects within images without relying on predefined categories, thus extending beyond the limitations of traditional object detection methods. Works such as GLIP [25] and Grounding DINO [34] have demonstrated the potential of grounding object detection in language, formulating detection as a problem of language grounding to learn instance-level visual representations with deep language-aware fusion. APE [48] combines detection and grounding into a unified model that can handle diverse tasks simultaneously. While previous work has significantly advanced open-set object detection, training large-scale multimodal detection frameworks still requires substantial computational power and data. Therefore, efficiently fine-tuning existing detection frameworks to better adapt to new downstream data presents a cost-effective solution. In this paper, we introduce multimodal inplace prompt tuning. This approach involves freezing the original model and fine-tuning a small number of added parameters to enhance detection performance for better adaptation.

### 2.2 Parameter-efficient Fine-tuning

By freezing most of the pretrained parameters and only updating a subset, models can achieve improved performance on downstream tasks [2, 3, 9, 18, 20, 29, 40, 49, 70, 72]. Adapter Tuning [14] have paved the way for more nuanced and efficient adaptation strategies with customized adapters. Since large language models are sensitive to prompt, Prefix-Tuning [27] and P-Tuning [35, 36] enable fine-tuning of models in a more efficient manner, focusing on the

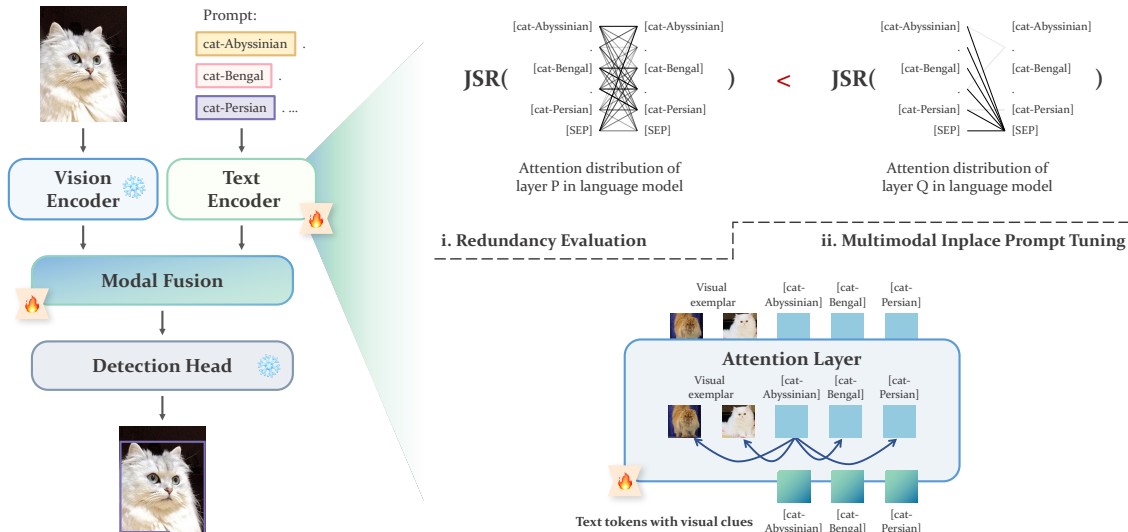

**Figure 2: An illustration of our proposed methods. To analyze the redundancy in the language model within the detector and precisely identify which parameters need fine-tuning, we consider both the variance within the attention distribution of the attention heads and between heads, based on the JSR. This approach helps to assess the level of redundancy in an attention layer. Then, by ranking the degrees of redundancy, we can accurately pinpoint the locations that require fine-tuning. After adding a small number of trainable parameters through LoRA in the language model, we concatenate pre-selected visual exemplars with the original category prompt as input. This allows the original language model to learn to refine the classification boundaries through visual exemplars via the MIPT approach.**

adaptation of prompt rather than the parameters within the model. The existing methods for adapting large language models, such as adapters, often result in unwanted side effects like inference latency, and the optimization of prompt tuning can be particularly challenging. LoRA [15] employs a low-rank decomposition approach to simulate parameter changes, where the updated weights of the trained low-rank matrices are simply added together to complete the update process. However, the wide variety of above methods and tasks makes choosing the appropriate one a challenge. UniPELT [41] further incorporates various methods into the transformer architecture. However, the fine-tuning methods mentioned above primarily involve adjustments from unimodality and do not concern the learning for multimodal data. To this end, our work aim to aggregate the multimodal information by tuning the text prompt with visual clues through low-rank adaptation.

### 2.3 Multimodal Framework With Visual Prompt

As model sizes increase, especially with large language models, a single set of weights is sufficient to tackle tasks across multiple modalities [5, 6, 12, 28, 33, 43, 51–53, 62, 63, 65, 68]. Inspired by this, MQ-Det [59] proposes modulated pre-training, enabling the detection models to utilize both textual and visual queries without extensive re-training. DINOv [22] delves deeper into visual in-context prompting for open-set segmentation, enabling the model to utilize provided bounding boxes and points as visual prompts. T-Rex2 [16] further addresses the challenge of long-tail categories in open-set detection through the use of visual in-context instructions.

Constructing detection boundaries for open-set detection often requires the assistance of large language models, but few work has been done to examine whether changes in these language model are adequately adapted to visual tasks. Our approach investigates the redundancy issue of language models in a multimodal framework. It utilizes visual clues to address the suboptimal encoding of text prompts caused by this redundancy.

## 3 METHOD

Training large-scale detection models from scratch incurs considerable computational and data costs [17, 33, 60]. Although previous studies have sought efficient strategies to enhance performance within existing detection frameworks [59, 67], there has been scant research on the suitability of language model modifications for detection tasks. As illustrated in the half bottom of Figure 2, our methods start by evaluating the positions of redundancy within the language model using Jensen-Shannon Redundancy (Sec. 3.2). Then, at the redundant positions, we introduce information from the visual counterpart by multimodal inplace prompt tuning(Sec. 3.3). Furthermore, we can use the features of visual regions and text embeddings after fusion module for further cross-modal distillation and alignment.(Sec. 3.4).

### 3.1 Preliminary

Our methods mainly based on the grounding paradigm detection framework [25, 34]. As illustrated on the top of Figure 2, images are

first encoded into visual tokens by a visual encoder (e.g. Swin Transformer [38], ViT [7]). Meanwhile, the text of the categories (e.g. cat-Abyssinian. cat-Bengal. cat-Persian...), linked together by periods ".", forms the prompt for detection (e.g. prompt: [cat-Abyssinian. cat-Bengal. cat-Persian]). Similar to image encoding, this prompt is fed into the language model (e.g. BERT [4], RoBERTa [37]) and encoded into text tokens. After implementing separate encodings for different modalities, deep fusion of visual and text features is essential for learning a high-performance phrase grounding model [23, 24, 26, 39, 50]. Through the modal fusion module, a deep integration between the image and language embeddings occurs, merging multimodal information in the final layers of encoding. By utilizing region-word alignment with contrastive learning, positive sample pairs of the same category are brought closer together, while negative ones are pushed further apart. Finally, the model learns how to perform classification and bounding box regression in detection head from a large amount of region-text pairs.

Examining the entire pipeline of open-set multimodal detection, large language models play vital role within it. World knowledge pre-trained in large language models aids in constructing effective classification boundaries. Via full-scale training, although it enables large language models with world knowledge to better align with detection tasks, it can lead to degradation of attention patterns. If used directly in downstream task inference, it might not perform well, especially for long-tail and fine-grained categories. Continuing to resolve this issue through full-model training would further consume substantial training resources. MQ-Det [59] is the first to integrate visual queries into an established language-only query detector. However, MQ-Det introduces visual information into text encoding through a cascaded attention module, then empirically determines the optimal placement of this new module within the language model. This hinders the effective utilization of added parameters and the cost of model inference. Our study addresses redundancy in language model detectors by using metrics to efficiently introduce learnable parameters and improve performance through inplace prompt tuning without raising costs.

## 3.2 Jensen-Shannon Redundancy (JSR)

We first introduce the Jensen-Shannon Redundancy (JSR). This novel approach aims to identify redundancy within the attention mechanisms of large language models. JSR applies the Jensen-Shannon divergence to evaluate and minimize redundancy within the attention mechanisms of large language models. This method quantifies divergence in attention distribution both within individual transformer heads (intra-head redundancy) and across different heads (inter-head redundancy). By pinpointing where attention patterns are overly uniform or duplicated, it identifies the most redundant parts of the model.

*3.2.1 Intra-head Redundancy.* Intra-head Redundancy is calculated to measure the redundancy of attention distributions within each individual head of a transformer layer. Given a head $h$ with token length $L$, let $D_{i,h}$ represent the attention distribution for the $i$-th token, corresponding to the sequence positions. The average attention distribution $\bar{D}_h$ for head $h$ is computed as the mean of

the attention distributions across all token positions:

$$\bar{D}_h = \frac{1}{L} \sum_{i=1}^{L} D_{i,h}. \tag{1}$$

Subsequently, the Jensen-Shannon divergence $JS(D_{i,h}||\bar{D}_h)$ is calculated for each token distribution $D_{i,h}$ relative to the average distribution $\bar{D}_h$. The intra-head redundancy $U_{\text{intra},h}$ for head $h$ is then defined as the mean Jensen-Shannon divergence across all tokens, which is:

$$U_{\text{intra},h} = \frac{1}{L} \sum_{i=1}^{L} JS(D_{i,h}||\bar{D}_h). \tag{2}$$

*3.2.2 Inter-head Redundancy.* Inter-head Redundancy evaluates the consistency of attention distributions across the different heads within the same layer. For each token position $i$, the average attention distribution $\bar{D}_i$ across all heads $H$, is calculated as:

$$\bar{D}_i = \frac{1}{H} \sum_{h=1}^{H} D_{i,h}. \tag{3}$$

The Jensen-Shannon divergence $JS(D_{i,h}||\bar{D}_i)$ is then computed for the distribution of each head $h$ compared with the average distribution $\bar{D}_i$. The inter-head uniformity $U_{\text{inter}}$ for the layer is determined by averaging these divergences across all token positions, yielding:

$$U_{\text{inter}} = \frac{1}{L} \sum_{i=1}^{L} \frac{1}{H} \sum_{h=1}^{H} JS(D_{i,h}||\bar{D}_i). \tag{4}$$

The final metric for assessing the importance of a layer, $I_{\text{layer}}$, is computed as a sum of the weighted intra-head uniformity for all heads and the inter-head uniformity, adjusted by a hyperparameter $\lambda$ to ensure both components contribute on a comparable scale:

$$R_{\text{layer}} = \lambda \cdot \sum_{h=1}^{H} U_{\text{intra},h} + U_{\text{inter}}, \tag{5}$$

where $\lambda$ is selected to balance the contributions of intra-head and inter-head redundancy, facilitating a comprehensive assessment of the significance of each layer within the language model.

By employing the JSR, we effectively pinpoint superfluous attention patterns and determine the optimal subset of parameters for fine-tuning. This strategy not only reduces the computational overhead but also ensures that performance enhancements are substantial by strategically targeting areas of the model that contribute the least to information diversity. Further details will be elaborated in the experimental section.

## 3.3 Multimodal Inplace Prompt Tuning (MIPT)

Building upon our proposed Jensen-Shannon Redundancy metric, the next stage of model refinement is implemented through Multimodal Inplace Prompt Tuning. This process aims to enrich the model's text prompts with visual context, thereby addressing the challenge of redundant attention patterns and enhancing prompt representations inplace for improved object detection in open-world scenarios.

**Algorithm 1:** PyTorch-style Pseudocode for MIPT

```
(1)  # Initialization of MIPTSelfAttention
(2)  class MIPTSelfAttention(BertSelfAttention):
(3)      def __init__(self, config):
(4)          super().__init__(config)
(5)          self.register_parameter("vision_gate", nn.Parameter(torch.zeros(1)))
(6)          self.Q.add_lora()
(7)          self.V.add_lora()
(8)      # Forward pass with text and vision inputs
(9)      def forward(self, text, vision, text_attention_mask, vision_attention_mask):
(10)         # The total number of visual exemplars in the vision clues bank
(11)         vision = self.project_and_norm(vision)
(12)         num_vision = vision_attention_mask.shape[-1]
(13)         hidden_states = torch.cat((vision, text), dim=1)
(14)         # QKV operations
(15)         query_layer, key_layer, value_layer = self.QKV(hidden_states)
(16)         attention_scores = calc_attention_score(query_layer, key_layer)
(17)         # Separate visual and textual information
(18)         vision_value_layer = value_layer[:, :, :num_vision, :]
(19)         text_value_layer = value_layer[:, :, num_vision:, :]
(20)         vision_attention_scores = attention_scores[:, :, num_vision:, :num_vision]
(21)         text_attention_scores = attention_scores[:, :, num_vision:, num_vision:]
(22)         # Calculate context layer by weighting value layer
(23)         text_context_layer = calc_context(text_attention_scores, text_attention_mask, text_value_layer)
(24)         vision_context_layer = calc_context(vision_attention_scores, vision_attention_mask,
                 vision_value_layer)
(25)         # Modal merge using vision gate
(26)         context_layer = torch.tanh(self.vision_gate) * vision_context_layer + text_context_layer
(27)         # Reshape and return output
(28)         outputs = (context_layer, text_attention_scores) if output_attentions else (context_layer,)
(29)         return outputs
```

### 3.3.1 Vision Clues Preparation.
Following MQ-Det [59], we expand the coordinates of all bounding boxes by a factor of 1.5, allowing us to capture the contextual backdrop of each target object. Subsequently, the raw images are processed through a visual encoder and a FPN [32] to extract multi-scale feature representations. ROI Align [46] is then employed to project the expanded bounding boxes onto the corresponding feature scale, extracting the target features from the feature map. These extracted object features are cataloged, forming a comprehensive visual clue bank that serves as a reservoir of candidates for subsequent utilization in our MIPT framework.

### 3.3.2 Inplace Prompt Tuning via Visual Clues.
Leveraging JSR, next step begins with a thorough evaluation of redundancy across various layers of the language model. This assessment facilitates an ordered layer selection based on the degree of redundancy, identifying those that are most amendable to enhancement, as shown in Figure 4. Subsequently, we freeze the attention parameters of the original redundant layers. As shown in the line 6 and 7 of the Algorithm 1, a LoRA[15] bypass is then integrated into the query and value linear layers of the attention mechanism. The mathematical

expression is as follows:

$$W = W_0 x + \Delta W x = W_0 x + BA x \quad (6)$$

In equation 6, $W_0$ represents the frozen pretrained weight matrix of the model. $B$ and $A$ are the trainable rank decomposition matrices, specifically introduced during the adaptation process. $x$ denotes the input to the weight matrix. By refining the model in this fashion, we facilitate the language model's ability to construct nuanced classification boundaries enriched with visual context. This not only streamlines the integration process but also preserves computational efficiency, as the core structural elements of the language model are left unaltered, thereby avoiding the need for additional inference modules.

The core of Multimodal Inplace Prompt Tuning lies in the fusion of self-attention results (context layers) derived from both visual and textual streams. This fusion is guided by the learnable vision gate $\gamma$, initialized in the line 5 of the Algorithm 1, which modulates the influence of visual information on the merged attention result (context layer).

During the forward pass, the process starts by projecting and normalizing visual inputs. Visual counterparts are first passed through

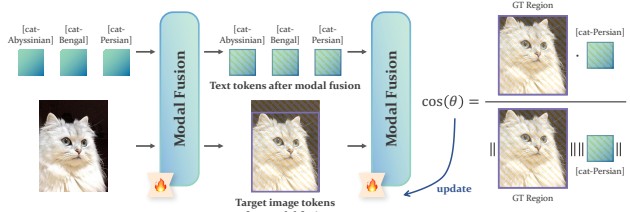

**Figure 3: An illustration of Cross modal Self-distillation. After the transformation through the fusion block, the image features and text features are further enhanced in terms of relevant information within each modality, while irrelevant information is diminished. Through Cross modal Self-distillation, the LoRA module is adjusted to allow the fusion block to learn to finely discern areas of high relevance in the target images based on the detached text features.**

a linear transformation and normalization process. Specifically, given a set of visual embeddings, we apply a linear transformation to match the dimensionality of the text embeddings. This transformed visual information is then normalized to ensure stability and consistency in the scale of features, as shown in the line 11. The process is followed by concatenating these visual features with textual inputs to create a unified feature set, as shown in line 13. The purpose of this is to keep the computational process of the original text encoding unchanged. This combined feature set is then processed through a modified attention mechanism, first by performing Query, Key, Value (QKV) operations, as shown in line 15, and then by calculating attention scores to establish relevancies across the concatenated multimodal data, as shown in line 16. Distinctly, this method separates visual and textual features, as illustrated from the line 18 to line 21 in the Algorithm 1, computes attention results for each modality by applying their respective attention scores. The textual self-attention results ($C_T$), as shown in the line 23, is obtained directly from the textual stream's attention mechanism, without further modification. This preserves the integrity and richness of textual information, serving as a robust foundation for subsequent fusion. Meanwhile, to obtain the visual self-attention results ($C_V$), the attention of the text token will be calculated to assign the weight of several visual examples of the same class, thus weighting these visual examples to obtain a visual counterpart $C_V$ optimized for this text token, as shown in line 24.

A pivotal aspect of this method is inplace tuning for the visual self-attention results using the shared attention from the text, which is achieved through the application of Low-Rank Adaptation [15] to fine-tune the existing attention weights. The modal merge operation, as shown in line 26, computes the final modal-merged results $C$ by combining both modalities:

$$C = \tanh(\gamma) \cdot C_V + C_T \tag{7}$$

This equation illustrates how the model dynamically adjusts the contribution of visual information, ensuring that the final attention result leverages complementary cues from both modalities. The

addition of $C_T$ directly, without scaling, ensures that textual information retains its foundational role, while $\tanh(\gamma) \cdot C_V$ introduces a modulated visual perspective.

## 3.4 Cross modal Self-distillation (CMSD)

Continuing with MIPT, we further delve into the Cross modal Self-distillation method. Take the fusion module of GLIP for example, which contains three submodules:

$$I_{i2t}^i, T_{i2t}^i = \text{CrossAttn}(I^i, T^i), \quad i \in \{0, 1, \dots, L-1\} \tag{8}$$

$$I^{i+1} = \text{ImageEncoderLayer}(I^i + I_{i2t}^i), \tag{9}$$

$$T^{i+1} = \text{TextEncoderLayer}(T^i + T_{i2t}^i), \tag{10}$$

where $L$ is the number of fusion block, $CrossAttn$ denotes the fusion operation for text and image. $ImageEncoderLayer$ takes the image features $I^i$ and image features reconstructed from text $I_{t2i}^i$ as inputs. In a similar way, $TextEncoderLayer$ takes the text features $T^i$ and text features reconstructed from image $T_{i2t}^i$ as inputs. CMSD uses the output from $CrossAttn$ as input for the cosine similarity loss, with the goal of further aligning the semantic gap between image regions and category prompts. Concretely, as depicted in Figure 3, firstly, text tokens and corresponding target image undergo $CrossAttn$ fusion, where their features are combined. After fusion, the text features are detached from the computation graph to serve as targets, and then, based on the cosine distance, the gap between the image features and the text features is narrowed. We freeze the original parameters of $CrossAttn$ in the fusion block and then introduce a trainable LoRA module to encourage $CrossAttn$ to better distill cross-modal knowledge. This optimizes the congruence between the text token and corresponding ground truth region. This is mathematically expressed as:

$$\text{Align} = \frac{1}{N_{\text{num\_gt}}} \sum_{i=1}^{N_{\text{num\_gt}}} \left(1 - \text{CosSim}(I^{i+1}, T^{i+1})\right) \tag{11}$$

In this formula, $N_{\text{num\_gt}}$ denotes the number of ground truth regions in the image, $I^{i+1}$ symbolizes the pooled visual features, and $T^{i+1}$ represents the targeted text embeddings. The Cross modal Self-distillation seeks to minimize this alignment loss, ensuring that the visual features are finely tuned to reflect the textual data, thus fostering a deep, semantic synergy between the visual and textual information. This process improves the model's ability to recognize and categorize objects based on both visual and textual cues, bridging the gap between the abstract semantic knowledge in language models and the concrete visual information captured by the detection system.

## 4 EXPERIMENT

### 4.1 Experimental Setup

*4.1.1 Benchmarks.* In our research, we have assessed the effectiveness of our proposed methods across two distinctive benchmarks to demonstrate their robustness and versatility in object detection tasks.

The first benchmark utilized is the LVIS dataset [11], which contains a wide array of detection targets spanning more than 1200 categories. This dataset is notably leveraged for evaluating detection outcomes on long-tail distributed data. Consistent with

| Model | Backbone | LVIS MiniVal (%) | | | | LVIS Val v1.0 (%) | | | | ODinW-13 (%) | ODinW-35 (%) |
|---|---|---|---|---|---|---|---|---|---|---|---|
| | | AP | $AP_r$ | $AP_c$ | $AP_f$ | AP | $AP_r$ | $AP_c$ | $AP_f$ | $AP_{avg}$ | $AP_{avg}$ |
| MDETR [17] | R101 | 24.2 | 20.9 | 24.9 | 24.3 | 22.5 | 7.4 | 22.7 | 25.0 | 25.1 | 10.7 |
| GLIP-T [25] | Swin-T | 26.0 | 20.8 | 21.4 | 31.0 | 17.2 | 10.1 | 12.5 | 25.5 | 41.9 | 18.7 |
| GLIP-L [25] | Swin-L | 37.3 | 28.2 | 34.3 | 41.5 | 26.9 | 17.1 | 23.3 | 35.4 | 51.0 | 22.6 |
| GLIPv2-T [67] | Swin-T | 29.0 | - | - | - | - | - | - | - | 50.7 | 22.3 |
| G-DINO-T [34] | Swin-T | 25.7 | 15.2 | 21.9 | 30.9 | - | - | - | - | 49.8 | 21.7 |
| BARON [56] | R50 | - | - | - | - | 29.5 | 23.2 | 29.3 | 32.5 | - | - |
| OWL-ViT [42] | ViT-L | - | - | - | - | 34.6 | - | - | - | - | 18.8 |
| MQ-GLIP-T [59] | Swin-T | 30.4 | 21.0 | 27.5 | 34.6 | 22.6 | 15.4 | 18.4 | 30.4 | 45.6 | 20.8 |
| MQ-GLIP-L [59] | Swin-L | 43.4 | 34.5 | 41.2 | 46.9 | 34.7 | 26.9 | 32.0 | 41.3 | 54.1 | 23.9 |
| G-DINO-T (MIPT) | Swin-T | 29.5 | 17.4 | 25.5 | 35.2 | 21.9 | 11.2 | 17.3 | 31.7 | 49.7 | 21.9 |
| GLIP-T (MIPT) | Swin-T | 30.4 | 20.7 | 26.9 | 35.1 | 23.1 | 17.8 | 18.4 | 30.5 | 47.3 | 21.8 |
| GLIP-L (MIPT) | Swin-L | **45.0** | **36.9** | **42.8** | **48.5** | **35.9** | **28.7** | **32.9** | **42.3** | **54.1** | **25.0** |

Table 1: Directly transferred evaluation on multiple benchmarks. "–" indicates that the work does not have a reported number.

Figure 4: An illustration of redundancy evaluation via JSR.

GLIP, for directly transferred evaluation, owing to its extensive category range where not all classes can be included in a single text prompt, we segmented the categories into multiple prompts. Each prompt encapsulated 40 classes, and the model was queried multiple times using varying prompts. The performance was reported on both the LVIS Minival subset, comprising 5000 images, and the comprehensive LVIS V1.0 validation set.

The second benchmark is the ODinW dataset [21], which aggregates 35 public object detection datasets to evaluate the comprehensive performance of pretrained large-scale detection models. To further evaluate core model performance, we utilized its less noisy subset, ODinW-13, which provides a concentrated measure of a model's fundamental detection capabilities.

*4.1.2 Implementation details.* In the implementation details of our experiments, we employed GLIP [25] and G-DINO [34] as our baseline detectors. Firstly, through the Jensen-Shannon Redundancy analysis, we pinpointed the layers with the highest redundancy within the large language models, selecting the top six for further optimization. For instance, as illustrated in the Figure 4, we utilized Jensen-Shannon Redundancy to assess the redundancy of the 12 attention layers in the BERT[4] within GLIP-T. Based on these

results, we selected the 7-12 layers of the BERT model for subsequent fine-tuning. We then incorporated the LoRA module (via MIPT) serving as tunable parameters to adjust original attention weights for different modalities. Pre-extracted visual exemplars were utilized. For each category, five random images were chosen and processed through the vision encoder and ROIPool [46] to extract specific regional features [59]. For pre-training on Object365 [47], we conducted a single epoch with all original model parameters frozen, adjusting only the newly introduced parameters. During this stage, the learning rate for LoRA was set to 0.00001, and the learning rate for the gates was 0.0005. The pre-training utilized 8 V100 GPUs for GLIP-T and G-DINO-T, and 16 V100 GPUs were employed for GLIP-L.

## 4.2 Main Results

*4.2.1 Comparison with The Baselines.* In the comparison of our methods, we benchmarked against several leading approaches in open-vocabulary detection as shown in the Table 1, including MDETR [17], GLIP [25], G-DINO [34], BARON [56], and OWL-ViT [42], which utilize various backbones like ResNet101 [13], Swin Transformers [38], and Vision Transformers [7]. Our experiments revealed that the MIPT-enhanced versions of G-DINO-T, GLIP-T, and GLIP-L significantly outperformed their original counterparts. For example, G-DINO-T (MIPT) improved its Average Precision (AP) on the ODinW-13 benchmark by 3.8%, and GLIP-T (MIPT) showed an increase of 4.4% on the LVIS MiniVal dataset. Most notably, GLIP-L (MIPT) demonstrated exceptional performance, not only in comparison with its baseline model—achieving an AP increase of 7.7% on LVIS MiniVal—but also outshining all other models across every benchmark dataset, including the highest AP on ODinW-35 at 25.0%. This comprehensive performance uplift underscores the advantage of the MIPT framework in enhancing detection systems. MIPT's fine-tuning of under 1% of the model's parameters underscores the framework's efficiency, achieving state-of-the-art results with minimal parameter increase, a testament to the efficacy of incorporating visual clues and targeted parameter updates.

*4.2.2 Few-shot Tuning.* Furthermore, we conducted a comparative study on the effectiveness of few-shot tuning. As shown in the Table 2, GLIP-T model fine-tunes across all its parameters, totaling 233.4 million, which is resource-intensive and less adaptable to rapid deployment scenarios. In contrast, our MIPT-enhanced GLIP-T involves fine-tuning less than 1% of the total parameters, a stark reduction that still results in superior performance. In the 3-shot and 5-shot settings on the ODinW-13 benchmark, GLIP-T (MIPT) not only demonstrates a notable increase in accuracy (57.2% and 58.2%, respectively) but does so with an incredibly marginal trainable parameters (only 0.8% compared with original GLIP-T). This illustrates the efficiency of MIPT in achieving high precision in object detection with minimal computational overhead, making it particularly suitable for applications requiring quick adaptation to new environments with limited examples.

| Model | Trainable params | ODinW-13 (%) | |
|---|---|---|---|
| | | 3-shot | 5-shot |
| GLIP-T [25] | 233.4M | 54.9 | 56.4 |
| GLIP-T (MITP) | 1.94M | 57.2 | 58.2 |

**Table 2: Comparison of GLIP-T (fully-finetuned) and GLIP-T with Multimodal Inplace Prompt Tuning (MITP) on the ODinW-13 benchmark in 3-shot and 5-shot settings, showing performance improvements with minimal parameter increase.**

*4.2.3 Visualization of Redundant Attention Mechanisms.* The visualizations shown in the Figure 5 represent the attention mechanisms within certain heads of a transformer model, specifically focusing on the redundancy of these mechanisms. Each vertical panel depicts the attention distribution from one particular head (e.g., Head 0-6 denotes head 6 in layer 0, Head 3-11 denotes head 11 in layer 3, etc.) to prompt concatenated by various category tokens such as 'ball', 'ballet skirt', and 'banana'. The strength and pattern of attention, indicated by the thickness and the number of lines connecting tokens, highlight the model's focus. For instance, denser clusters suggest concentrated attention, which could signal redundancy if such patterns do not contribute to diverse semantic understanding. These visualizations are critical for identifying and rectifying redundant attention, thereby improving the model's efficiency and accuracy. This process forms an integral part of optimizing language models for detection tasks, as explored in our experiments.

*4.2.4 Ablation Study.* Our ablation study investigates the individual and combined effects of Jensen-Shannon Redundancy (JSR), Cross-modal Self-distillation, and Multimodal Inplace Prompt Tuning (MIPT) on the performance of our object detection model. The study's findings, illustrated in the Table 3, clearly demonstrate the incremental improvements achieved by integrating these components into our framework. Initially, without any of these techniques, the model (original GLIP-T) achieves baseline performances of 41.9% on ODinW-13 and 26.0% on LVIS MiniVal. Implementing MIPT to the first 6 layers of BERT improves these scores to 43.2%

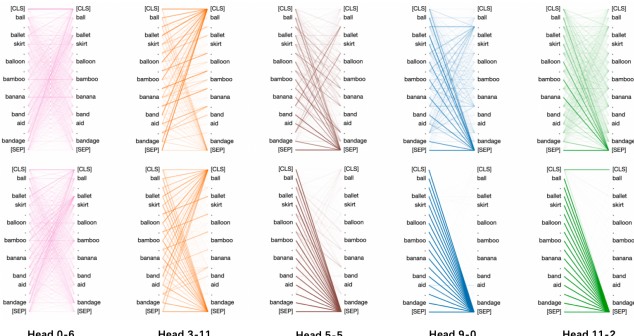

**Figure 5: illustrates the difference between a pretrained BERT model(the first row) and the BERT model within GLIP that has been fine-tuned for a detection task(the second row). It indicates that in GLIP, the diversity of the heads in BERT gradually diminishes as a result of subsequent fine-tuning.**

and 29.4%, respectively. The addition of JSR further enhances performance, indicating its effectiveness in reducing redundancy and focusing model training on impactful features. Combining all three techniques yields the best results, pushing the scores to 47.3% on ODinW-13 and 30.4% on LVIS MiniVal, validating our approach in enhancing detection accuracy through strategic component integration.

| JSR | Distillation | MIPT | ODinW-13 (%) | LVIS MiniVal (%) |
|---|---|---|---|---|
| ✗ | ✗ | ✗ | 41.9 | 26.0 |
| ✗ | ✗ | ✓ | 43.2 | 29.4 |
| ✓ | ✗ | ✓ | 46.8 | 30.1 |
| ✓ | ✓ | ✓ | 47.3 | 30.4 |

**Table 3: Ablation on Jensen-Shannon Redundancy, Cross modal Self-distillation and Multimodal Inplace Prompt Tuning**

# 5 CONCLUSION

In this paper, we present a novel strategy for enhancing open-set object detection by integrating large language models with multimodal inplace prompt tuning Framework. Our approach starts with the utilization of Jensen-Shannon Redundancy (JSR) to pinpoint and mitigate redundant attention patterns in language models adapted for detection tasks. This targeted fine-tuning improves computational efficiency and detection effectiveness. Further incorporation of MIPT, which enriches text prompts with visual clues, significantly boosts performance on the LVIS and ODinW datasets, demonstrating notable accuracy and adaptability improvements, especially in few-shot scenarios. Our findings highlight the transformative potential of combining metric-based redundancy evaluations with multimodal fine-tuning to optimize large language models for complex detection frameworks. Future work could expand this approach to include more modalities and applications, propelling further advancements in multimodal learning and object detection.

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
