# OpenReview forum: "Multimodal Inplace Prompt Tuning for Open-set Object Detection"
_acmmm.org/ACMMM/2024/Conference — MM2024 Poster_

### Official Review · Reviewer_gGAU · 2024-05-23

**Rating:** 5
**Confidence:** 2

**Summary:**

This paper introduces a novel framework called Multimodal Inplace Prompt Tuning (MIPT) for enhancing open-set object detection by integrating large language models (LLMs) with detection systems. The primary focus is on addressing the redundancy in attention patterns of LLMs when applied to object detection tasks. The authors propose a Jensen-Shannon Redundancy (JSR) metric to identify and mitigate redundant attention patterns. By incorporating visual clues into text prompts through MIPT, the study demonstrates significant improvements in detection performance across various benchmarks. Experimental results highlight substantial performance gains with minimal parameter tuning, showcasing the framework's efficiency and effectiveness.

**Strengths:**

Innovative Approach: The introduction of the Jensen-Shannon Redundancy (JSR) metric and Multimodal Inplace Prompt Tuning (MIPT) is innovative, providing a new method to enhance the alignment of language and visual features in detection tasks.

Performance Improvement: The experimental results demonstrate significant improvements in detection accuracy across multiple benchmarks, such as GLIP-L’s performance increase on ODinW-35 and LVIS.

Parameter Efficiency: The proposed method achieves notable performance gains with less than 1% of the model's parameters being fine-tuned, highlighting the efficiency of the approach.

**Limitations:**

While the paper includes ablation studies, more detailed experiments isolating the effects of individual components (such as different layers and heads in the LLM) would strengthen the claims.

**Suitability:**

2

---

### Official Review · Reviewer_rQ4v · 2024-05-24

**Rating:** 5
**Confidence:** 2

**Summary:**

The first to discover that directly fine-tuning language models in detection systems results in redundant attention patterns and leads to suboptimal prompt representations. Thus, in turn, falls to detect such fine-grained or long-tail categories, as shown in their figures. To this end, given the language model adaptation for detection that has been previously overlooked, we introduce a novel Multi-modal Inplace Prompt Tuning Framework (MIPT) for open-set object detection. In this paper, it first identify redundancy in attention distribution when integrating large language models into detection frameworks. It introduces the JSR metric to efficiently evaluate and pinpoint attention pattern redundancies in language models during fine-tuning for detection tasks.  The MIPT works by recalibrating prompt representations using visual cues for language models.

**Strengths:**

1. This paper is the first to discover that directly fine-tuning language models in detection systems results in redundant attention patterns which is novel and interesting and it provides empirical experiments on showing this phenomenon. The introduction of the Jensen-Shannon Redundancy metric to measure the redundant attention is convincing.

2. The experiment results outcomes on GLIP-T, GLIP-L, and G-DINO-T showcase the remarkable efficacy of the MIPT framework, yielding substantial performance enhancements which indicates the good performance of MIPT.

3. The part for Visualization of Redundant Attention Mechanisms is very interesting and this contributes for further research.

**Limitations:**

In the figure 4 and figure 5, the author illustrates the redundancy evluation in different layer. Would there exist a way in visualizeing the MIPT can reduce this redundancy?

**Suitability:**

3

---

### Official Review · Reviewer_kBwz · 2024-05-25

**Rating:** 4
**Confidence:** 3

**Summary:**

This paper investigates how to efficiently leverage LLMs for fine-tuning in order to enhance open-set object detection.  The contribution of this work is introducing LoRA to this task. It introduces a method to assess redundancy within language models using Jensen-Shannon Redundancy and proposes a multimodal inplace prompt tuning approach to improve prompt representations in detection systems. Experimental results show the  the effectiveness of the proposed framework.

**Strengths:**

1. Compared to full training, the proposed method can improve detection performance by fine-tuning a small number of parameters through LoRA.
2. The writing in the paper is clear and easy to understand.
3. The experimental results show performance improvements.

**Limitations:**

1. The pseudocode in the paper is overly complicated and lengthy. I suggest the authors use an abstract algorithm instead to facilitate research and analysis.
2. One of the main contributions of the paper is introducing LoRA, a fine-tuning method for LLMs, into multimodal scenarios to enhance the visual open-set object detection task. However, more analysis and ablation studies should be provided on the design for reducing fine-tuning parameters in this context to illustrate the value of this contribution.
3. Lack of discussion on the potential limitations of the proposed method, which is crucial as it contributes to a comprehensive understanding of the method.

**Suitability:**

2

---

### Meta-Review · Area_Chair_qJJv · 2024-07-08

**Recommendation:** Accept (Poster)
**Confidence:** 4

**Metareview:**

The paper introduces a novel framework called Multimodal Inplace Prompt Tuning (MIPT) for enhancing open-set object detection by integrating large language models (LLMs) with detection systems. The primary focus is on addressing redundancy in attention patterns of LLMs when applied to object detection tasks. Reviewers appreciated the innovative introduction of the Jensen-Shannon Redundancy (JSR) metric and the significant performance improvements demonstrated through experiments. However, they noted the pseudocode's complexity, the need for more detailed ablation studies, and a lack of discussion on potential limitations. Overall, the reviewers leaned towards acceptance, recognizing the framework's efficiency and effectiveness while suggesting areas for further refinement.